# Determinants of SARS-CoV-2 Infection in the Older Adult Population: Data from the LOST in Lombardia Study

**DOI:** 10.3390/vaccines10070989

**Published:** 2022-06-22

**Authors:** Giansanto Mosconi, Chiara Stival, Alessandra Lugo, Carlo Signorelli, Andrea Amerio, Luca Cavalieri d’Oro, Licia Iacoviello, David Stuckler, Alberto Zucchi, Anna Odone, Silvano Gallus

**Affiliations:** 1Department of Public Health, Experimental and Forensic Medicine, University of Pavia, 27100 Pavia, Italy; giansanto.mosconi01@universitadipavia.it; 2Department of Environmental Health Sciences, Istituto di Ricerche Farmacologiche Mario Negri IRCCS, 20156 Milan, Italy; chiara.stival@marionegri.it (C.S.); alessandra.lugo@marionegri.it (A.L.); silvano.gallus@marionegri.it (S.G.); 3Department School of Medicine, University Vita-Salute San Raffaele, 20132 Milan, Italy; signorellicarlo2307@gmail.com; 4Department of Neuroscience, Rehabilitation, Ophthalmology, Genetics, Maternal and Child Health (DINOGMI), Section of Psychiatry, University of Genoa, 16132 Genoa, Italy; andrea.amerio@unige.it; 5IRCCS Ospedale Policlinico San Martino, 16132 Genoa, Italy; 6Agenzia di Tutela della Salute della Brianza, 20900 Monza, Italy; luca.cavalieridoro@ats-brianza.it; 7Research Center in Epidemiology and Preventive Medicine (EPIMED), Department of Medicine and Surgery, University of Insubria, 21100 Varese, Italy; licia.iacoviello@uninsubria.it; 8Department of Epidemiology and Prevention, IRCCS Neuromed, 86077 Pozzilli, Italy; 9Department of Social Sciences and Politics, Bocconi University, 20136 Milan, Italy; david.stuckler@unibocconi.it; 10Agenzia di Tutela della Salute di Bergamo, 24121 Bergamo, Italy; alberto.zucchi@ats-bg.it

**Keywords:** determinants, infection, SARS-CoV-2, COVID-19, older adults, chronic diseases, older age

## Abstract

Most COVID-19 fatalities have occurred among older adults; however, evidence regarding the determinants of SARS-CoV-2 infection in this population is limited. Telephone interviews were conducted in November 2020 with a representative sample of 4400 Italians aged ≥65 years from the Lombardy region. We determined the prevalence of a history of SARS-CoV-2 infection. Through unconditional multiple logistic regression models, we estimated the odds ratios (ORs) of infection and the corresponding 95% confidence intervals (CIs). We further evaluated whether infection was related to a reduction in mental wellbeing. Of the participants, 4.9% reported a previous infection. No significant relationship between sex and infection was observed. Prior infection was less frequently reported in subjects aged ≥70 (OR = 0.55; 95% CI: 0.41–0.74) compared to 65–69 years, with no trend after 70 years of age. Those with at least one chronic condition reported a lower infection rate compared to healthy subjects (OR = 0.68; 95% CI: 0.49–0.93). Participants who lived alone more frequently reported infection than those who cohabited (OR = 2.33; 95% CI: 1.29–4.20). Prior infection was related to increased depressive symptoms (OR = 1.57; 95% CI: 1.17–2.10). This representative study of people aged ≥65 years suggests that in Italy, the oldest subjects and chronic patients less frequently exposed themselves to SARS-CoV-2 infection.

## 1. Introduction

During the first wave of COVID-19 contagion, Italy’s Lombardy region faced one of the earliest large outbreaks in Western countries, experiencing high levels of hospital congestion and a considerable COVID-19 fatality rate [1,2,3]. By the end of 2020, this region had registered the highest number of infections in the country, as well as one of the highest cumulative incidence rates of infection [1,4].

According to the national-level COVID-19 surveillance system, in Italy, by the end of 2020 (before vaccines were available), 96% and 86% of COVID-19-related fatalities had occurred among people aged 60 and 70 years and over, respectively [4]. Indeed, it is well documented in the literature that advanced age, being male, and the presence of chronic diseases are risk factors for adverse outcomes in COVID-19 patients [5,6].

Nonetheless, to date, knowledge about the potential determinants of SARS-CoV-2 infection in this population is mostly based on case series (which are inappropriate for investigating the issue as they do not include a comparison group) or surveys enrolling convenience samples (which provide findings that are not extensible to the general population) [7]. The few representative cross-sectional studies were online surveys, which are unsuitable for examining populations, such as the elderly, that often do not have internet access or have a low level of computer skills [8].

Within the Lockdown and lifeSTyles in Lombardia (LOST in Lombardia) project, this study aims to assess the prevalence of individuals self-reporting a history of SARS-CoV-2 infection and, above all, to investigate its possible determinants in a representative sample of Italian community-dwelling people aged 65 years or older.

## 2. Materials and Methods

A telephone-based cross-sectional survey was performed by Doxa, the Italian division of the Worldwide Independent Network/Gallup International Association, and coordinated by the Mario Negri Institute and other Italian universities and research institutions [9]. The LOST in Lombardia study was conducted between 17 and 30 November 2020 on a representative sample of 4400 community-dwelling older adults (aged 65 and over) from the Lombardy region (Northern Italy) [10].

Participants were randomly selected from a list of approximately 30,000 families, representative, by province and size of municipality, of families in the Lombardy region. A quota method was applied to ensure the representativeness of the sample in terms of sex, age, and province of residence. Overall, 10,365 older adults were contacted. Of them, 3705 people refused to participate and 2260 telephone calls were interrupted before the end of the questionnaire, while 4400 people completed the interview (response rate: 42%).

The study protocol was approved by the ethics committee (EC) of the coordinating group (EC of Fondazione IRCCS Istituto Neurologico Carlo Besta, File number 76, October 2020). All participants provided their informed consent to participate in the study.

Participants provided information regarding socio-demographic characteristics (such as sex, age, level of education, province of residence); marital status; and diagnosis of selected chronic conditions before pandemic onset (reference time: November 2019).

Anxiety and depressive symptoms were assessed using the two-item generalized anxiety disorder scale (GAD-2) [11] and the two-item patient health questionnaire (PHQ-2) [12], respectively. To evaluate any potential changes in psychological wellbeing, the presence of anxiety and depressive symptoms was assessed twice, referring to the time of the interview (November 2020) and one year before (November 2019).

Respondents reported if they had been infected with SARS-CoV-2 since the beginning of the pandemic and the type of diagnosis they received: a positive swab test, a positive serological test, or if they thought they had experienced signs or evident symptoms of COVID-19 without a confirmed diagnosis. We defined this measure as history of SARS-CoV-2 infection.

### Statistical Analysis

To investigate the determinants of SARS-CoV-2 infection, we estimated odds ratios (ORs) and their corresponding 95% confidence intervals (CIs) through unconditional multiple logistic models. As a sensitivity analysis, we re-ran all the models excluding subjects who reported only signs or symptoms but had not received a diagnosis. We defined a worsening in anxiety and depressive symptoms as any increase in the corresponding scales. All the models were adjusted for sex, age, level of education, province, and number of comorbidities. A statistical weight was applied to all the analyses to guarantee the representativeness of the sample in terms of sex, age, and province of residence.

All statistical analyses were performed using SAS 9.4 (Cary, NC, USA).

## 3. Results

### 3.1. Prevalence of a History of SARS-CoV-2 Infection

In November 2020, out of 4400 participants, 213 (4.9%) reported that they had been infected with SARS-CoV-2. Of the participants, 98 (2.2%) received a diagnosis through a positive swab test, 46 (1.1%) through a positive serological test, and 70 (1.6%) had experienced signs or symptoms without a confirmed diagnosis (Figure 1, Appendix B).

### 3.2. Determinants of SARS-CoV-2 Infection

Table 1 shows the ORs of subjects reporting SARS-CoV-2 infection by selected characteristics. No relationship was observed between sex and infection (compared to women, the OR for men was 1.02; 95% CI: 0.77–1.36). Subjects aged 70 years and older reported SARS-CoV-2 infection less frequently than those aged 65–69 years (OR = 0.55; 95% CI: 0.41–0.74). When considering participants aged 70 years or above, no trend of infection was observed with increasing age (p for trend = 0.447). Compared to participants from Milan (the Lombardy region’s capital city), participants from Bergamo and Brescia more frequently reported SARS-CoV-2 infection (OR = 2.11; 95% CI: 1.46–3.06). Divorced and single subjects more frequently reported that they had been infected compared to married subjects or those living with a partner (OR = 2.33; 95% CI: 1.29–4.20).

Compared to those with no diseases, subjects with one or more chronic conditions less frequently reported SARS-CoV-2 infection (OR = 0.68; 95% CI: 0.49–0.93). No trend was observed with an increasing number of chronic diseases (*p* for trend = 0.112). People with diabetes less frequently reported SARS-CoV-2 infection compared to subjects without this disease (OR = 0.64; 95% CI: 0.42–0.99), whereas people with migraine more frequently reported SARS-CoV-2 infection compared to those without this condition (OR = 2.01; 95% CI: 1.21–3.34; Table 2).

### 3.3. Worsening in Anxiety and Depressive Symptoms

Table 3 shows the OR of increased anxiety and depressive symptoms during the COVID-19 pandemic according to SARS-CoV-2 infection. No relationship was observed between infection and an increase in anxiety symptoms, whereas an increase in depressive symptoms was more frequently reported in individuals who contracted SARS-CoV-2 (OR = 1.57; 95% CI: 1.17–2.10). The sensitivity analysis produced comparable results.

## 4. Discussion

In November 2020, almost 5% of older people from the Lombardy region reported that they had experienced signs or symptoms compatible with COVID-19 or had tested positive for SARS-CoV-2 infection based on a swab or a serological test. No relationship between sex or level of education and SARS-CoV-2 infection was observed. Subjects aged 65–69 years more frequently reported that they had been infected compared to older subjects. Having one or more chronic diseases was related to a lower rate of infection. A SARS-CoV-2 infection was a determinant for increased depressive symptoms.

Based on our data, after almost one year from the beginning of the COVID-19 pandemic, 2.2% of adults aged 65 and older from the Lombardy region reported that they had received a positive swab. In November 2020, corresponding official data from the Italian National Health Institute showed that the cumulative incidence rate was around 3.5% in Italian adults aged 60 or older and around 3.8% in the Lombardy general population [13]. The lower prevalence of people who tested positive from a swab detected within our survey might be explained by the exclusion from our sample of individuals residing in nursing homes and subjects who died during 2020 due to their infection.

However, it is interesting to note that the percentage of participants who had been infected rose by 50% (reaching 3.3%) when we included those who had tested positive from a serological test and potentially more than doubled (up to 4.9%) when we included those who had experienced signs or symptoms compatible with COVID-19 without any test confirmation. These results might be explained by the fact that, in Italy, in the early stages of the pandemic, swabs tests were almost exclusively administered to patients in severe conditions, thus excluding a large number of individuals with no or mild signs or symptoms [14]. Accordingly, the Italian Ministry of Health conducted a seroprevalence study between 25 May 2020 and 15 July 2020, which revealed that 7.5% (95% CI: 6.8–8.3) of the Lombardy general population had SARS-CoV-2 antibodies [15], confirming the fact that the real cumulative incidence was underestimated when considering only swab data.

With regard to the determinants of infection, an important clarification must be made. Since COVID-19 is characterized by non-specific signs and symptoms [16], some of which can be very common in the elderly population [17,18], we conducted a sensitivity analysis by excluding subjects who reported only signs or symptoms without a confirmed diagnosis.

We did not observe any significant relationship between sex and SARS-CoV-2 infection, although our data seemed to suggest a marginally higher prevalence of infection in males. This is in line with a recent systematic review [19], which found that men have a slightly higher frequency of infection [20,21].

We found a lower frequency of infection in participants aged 70 years or older than in participants aged 65–69 years. This result could be explained by the fact that the latter group included a higher proportion of working subjects, who were more likely to be exposed to SARS-CoV-2 infection because of their more frequent social interactions. In addition, since social isolation increases with age [22], it is conceivable that subjects aged 70 years or older, who had fewer interactions, were better able to protect themselves from possible opportunities for infection. It is also likely that older people, being aware of the higher risks to which they were exposed, were more careful in complying with preventive measures [23]. Our findings are in line with a large systematic review and meta-analysis including 241 seroprevalence studies showing that the proportion of individuals aged 65 years or older with SARS-CoV-2 antibodies tended to decrease with age [20]. Furthermore, data from an Italian Ministry of Health survey show a decrease in seroprevalence in the over-70 age group compared to the 60–69 age group [15]. Since our survey did not allow for the inclusion of those who died from COVID-19, we recalculated the prevalence in every age group considering the number of expected cases, which was obtained by adding the proportion of deaths per age group based on the case fatality rate provided by the National Health Institute, referring to November 2020 [13]. The cases distribution by age remained substantially unchanged.

Participants residing in the neighboring provinces of Bergamo and Brescia showed a higher infection rate than participants from the Milan province. These results reflect data from seroprevalence studies in Lombardy [21,24]. In line with other studies [25,26], we found that divorcees and single persons reported SARS-CoV-2 infection more frequently than married or cohabiting individuals. However, the reasons for this finding are unclear.

The absence of chronic diseases was associated with a higher frequency of infection. This can be explained by the less cautious attitude of this sub-group compared to individuals with at least one chronic condition, who were probably aware of the increased risks they were facing. The presence of specific chronic diseases leads to a depression of the immune system [27] and an increased risk of COVID-19 resulting in adverse outcomes [5]. This has been shown particularly among patients with diabetes. In contrast, patients who reported that they had been diagnosed with some form of chronic migraine before the COVID-19 pandemic showed a higher infection rate.

Both COVID-19 and Post-COVID-19 Syndrome can occur with migraine [28,29]; however, our results seems to suggest that this condition could also result in an increased risk of infection. Further studies are needed to clarify this relationship.

A higher frequency of infection was observed in subjects reporting increased depressive symptoms during the pandemic. Evidence from the literature shows that people who suffered from COVID-19 frequently experienced a deterioration in their mental health [29,30,31,32]. Furthermore, people who reported being infected most likely had other cases in their family [33] and may have lost relatives as a result of COVID-19. The likelihood of reporting signs and symptoms attributable to COVID-19 may be related to the presence of depression and/or anxiety: studies comparing the prevalence of anxiety and depression among subjects who tested positive and subjects who tested negative to SARS-CoV-2 infection found different results [34,35], so further research is needed to understand the nature of this relationship.

Our findings should be interpreted in light of some limitations. The first is that data on diagnostic tests and COVID-19-like signs and symptoms were self-reported, which introduces potential recall bias. Furthermore, it must be noted that the testing capacity in Italy changed during 2020 and that some participants may have experienced an asymptomatic SARS-CoV-2 infection without ever being tested. It must also be taken into account that patients who die from COVID-19 frequently have worse basal clinical conditions compared to healed patients. Moreover, we did not have information on the severity of COVID-19 symptoms for people reporting SARS-CoV-2 infection. We also did not have information on the timing of the SARS-CoV-2 infection. This prevented us from analyzing the relationship with mental health outcomes according to the time since infection. More importantly, as this was a cross-sectional study, we could not establish any causal relationship, although every effort was made to provide a longitudinal context for the analyses. However, to the best of our knowledge, this is the first study to investigate the role of the possible determinants of SARS-CoV-2 infection in older people. We also believe that this study provides useful elements for obtaining a more reliable estimate of the magnitude of the cumulative incidence of COVID-19 in this population than data on swabs alone. Furthermore, we used a large representative sample of the population aged 65 or older from the Lombardy region, and data were collected through a telephone-based interview, which can be considered the most suitable solution for the older population in a pandemic context. During the pandemic, online surveys proved to be an effective tool for overcoming social-distancing measures [36]; however, they carry the risk of selection bias, since they exclude those who, like many seniors, have poor digital literacy and internet access [8].

## 5. Conclusions

Our results suggest that the oldest old and those with chronic conditions limited their exposure to SARS-CoV-2 infection. Although the COVID-19 fatality rate increases dramatically after the age of 70 [37], our study seems to indicate that those between the ages of 65 and 69 are more susceptible to SARS-CoV-2 infection and should be targeted by preventive strategies to better contain the spread of contagion.

## Figures and Tables

**Figure 1 vaccines-10-00989-f001:**
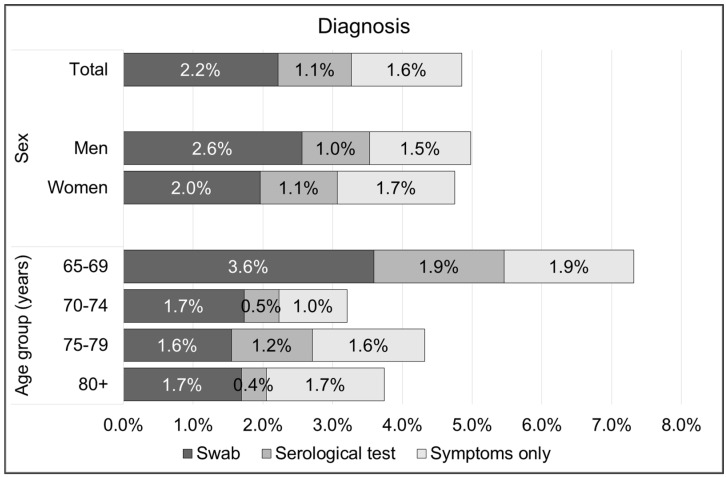
Distribution of 4400 Italian subjects aged 65 years or above according to COVID-19 infection and type of diagnosis, overall and by sex and age group. Lombardy region, Italy, 2020.

**Table 1 vaccines-10-00989-t001:** Distribution of 4400 Italian subjects aged 65 years or above according to SARS-CoV-2 infection, overall and by selected characteristics. Corresponding odds ratios ^ (OR) and 95% confidence intervals (CI). Lombardy region, Italy, 2020.

Characteristics	N	SARS-CoV-2 Infection Based on Confirmed Diagnosis or Symptoms		SARS-CoV-2 Infection Based on Confirmed Diagnosis Only *	
		%	OR (95% CI)	%	OR (95% CI)
Total	4400	4.9		3.3	
Sex					
Women	2498	4.7	1.00	3.1	1.00
Men	1902	5.0	1.02 (0.77–1.36)	3.6	1.11 (0.79–1.56)
Age group (years)					
65–69	1289	7.3	1.00	5.6	1.00
70–74	838	3.2	**0.45** (0.29–0.70)	2.3	**0.41** (0.24–0.70)
75–79	1188	4.3	**0.62** (0.43–0.88)	2.8	**0.52** (0.33–0.80)
80+	1085	3.8	**0.57** (0.38–0.86)	2.1	**0.41** (0.25–0.69)
P for trend			**0.006**		**<0.001**
Level of education					
Low	788	3.7	1.00	2.5	1.00
Intermediate	3168	5.2	1.20 (0.78–1.84)	3.5	1.04 (0.62–1.74)
High	444	4.3	0.92 (0.50–1.71)	3.1	0.81 (0.39–1.67)
P for trend			0.914		0.592
Province †					
MI	1424	3.5	1.00	2.4	1.00
CO + LC + MB + VA + SO	1262	4.8	1.34 (0.91–1.97)	3.2	1.24 (0.77–1.98)
BG + BS	1038	7.2	**2.11** (1.46–3.06)	4.6	**1.87** (1.19–2.95)
CR + LO + MN + PV	677	4.1	1.11 (0.69–1.79)	3.5	1.35 (0.79–2.31)
Marital status					
Married/living with a partner	3113	4.7	1.00	3.3	1.00
Divorced/separated	140	10.0	**2.33** (1.29–4.20)	5.4	1.66 (0.75–3.67)
Widowed	871	4.3	1.16 (0.77–1.73)	2.7	1.19 (0.73–1.95)
Never married	276	5.5	1.06 (0.61–1.84)	4.1	1.10 (0.58–2.08)

^ Estimated by unconditional multiple logistic regression models after adjustment for sex, age, education level, province, and number of chronic conditions; estimates in bold are those statistically significant at 0.05 level. * 70 subjects reporting COVID-19 symptoms but no diagnosis were excluded from the analyses. Reference category. † MI: Milan; CO: Como; LC: Lecco; MB: Monza and Brianza; VA: Varese; SO: Sondrio; BG: Bergamo; BS: Brescia; CR: Cremona; LO: Lodi; MN: Mantua; PV: Pavia.

**Table 2 vaccines-10-00989-t002:** Distribution of 4400 Italian subjects aged 65 years or above according to SARS-CoV-2 infection, overall and by chronic conditions. Corresponding odds ratios ^ (OR) and 95% confidence intervals (CI). Lombardy region, Italy, 2020.

Characteristics	N	SARS-CoV-2 Infection with Confirmed Diagnosis or Symptoms		SARS-CoV-2 Infection with Confirmed Diagnosis Only *	
		%	OR (95% CI)	%	OR (95% CI)
Total	4400	4.9		3.3	
Chronic diseases ^†^					
No	890	6.9	1.00	5.1	1.00
Yes	3510	4.3	**0.68 (0.49–0.93)**	2.9	**0.63 (0.43–0.92)**
1	1264	4.6	0.70 (0.48–1.01)	3.4	0.70 (0.45–1.08)
2	1375	3.8	**0.61 (0.41–0.90)**	2.4	**0.53 (0.33–0.85)**
3+	870	4.6	0.76 (0.50–1.17)	2.9	0.68 (0.40–1.14)
P for trend			0.112		**0.043**
Diabetes ^†^	936	3.2	**0.64 (0.42–0.99)**	2.2	0.71 (0.42–1.21)
Hypertension ^†^	2443	4.2	0.92 (0.63–1.35)	2.8	1.00 (0.63–1.59)
Osteoarthritis or arthritis ^†^	1485	4.6	1.18 (0.82–1.71)	3.1	1.39 (0.88–2.19)
Asthma ^†^	148	6.2	1.47 (0.73–2.97)	5.0	1.80 (0.81–3.98)
Cancer ^†^	170	5.8	1.50 (0.76–2.96)	1.9	0.70 (0.22–2.25)
Heart disease ^†^	539	4.3	1.07 (0.66–1.72)	2.6	1.05 (0.57–1.91)
Osteoporosis ^†^	652	4.6	1.10 (0.70–1.74)	2.8	1.05 (0.59–1.87)
Kidney failure ^†^	95	6.0	1.53 (0.63–3.75)	2.2	0.86 (0.20–3.65)
Bronchitis ^†^	188	4.4	0.96 (0.46–2.01)	2.9	0.96 (0.39–2.39)
Migraine ^†^	252	8.5	**2.01 (1.21–3.34)**	4.6	1.52 (0.78–2.98)

^ Estimated by unconditional multiple logistic regression models after adjustment for sex, age, education level, province, and number of chronic conditions; estimates in bold are those statistically significant at 0.05 level. * 70 subjects reporting COVID-19 symptoms but no diagnosis were excluded from the analyses. Reference category. ^†^ Diagnosed prior to the COVID-19 pandemic breakout.

**Table 3 vaccines-10-00989-t003:** Distribution of 4400 Italian subjects aged 65 years or above according to increased anxiety and depressive symptoms, by SARS-CoV-2 infection. Corresponding odds ratios ^ (OR) and 95% confidence intervals (CI). Lombardy region, Italy, 2020.

Characteristics	N	Increased Anxiety Symptoms during COVID-19 Pandemic		Increased Depressive Symptoms during COVID-19 Pandemic	
		%	OR (95% CI)	%	OR (95% CI)
SARS-CoV-2 infection with confirmed diagnosis or symptoms vs. no infection	213	39.9	1.13 (0.85–1.51)	37.8	**1.57 (1.17–2.10)**
SARS-CoV-2 infection with confirmed diagnosis only * vs. no infection or infection with no diagnosis	144	44.3	1.36 (0.97–1.90)	39.5	**1.68 (1.19–2.38)**

^ Estimated by unconditional multiple logistic regression models after adjustment for sex, age, education level, province, and number of chronic conditions; estimates in bold are those statistically significant at 0.05 level. * 70 subjects reporting COVID-19 symptoms but no diagnosis were excluded from the analyses.

## Data Availability

Data that support the findings of this study are available from the corresponding author, A.O., upon reasonable request. The data are not publicly available because some parts of the questionnaire have not yet been analyzed. As of June 2023, the data will be available and accessible to everyone without restriction.

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
