# Peer review of "Determinants of SARS-CoV-2 Infection in the Older Adult Population: Data from the LOST in Lombardia Study"

_vaccines, 2022, doi:10.3390/vaccines10070989_

Round 1
Reviewer 1 Report
This paper reports on the association of Covid 19 with characteristics of persons aged over 65 in Lombardy (Italy). There are several questions which should be answered before a possible publication.
1) 4400 persons are included and were contacted by phone. I guess that quite a few did not repond to the questions for different reasons. What was the percentage of persons not answering to the questions?
2) the PCR on swab is associated with recent infection, the immunological test with contact with the virus from some days to several weeks ago. Concerning anxiety or depression the timing of infection may be very important: I am anxious because I just got it and I am depressed because I got it weeks ago, since depression does not appear in few days. Even for other characteristics (age, marital status, etc..) the fact that you just become infected or you were infected may yeld different ORs. I suggest that the analysis should be done on swab and immunological tests separately.
3) A good part of the 4400 people declared having symptoms of Covid 19. I wonder if at this period flu was present or changes of temperature caused colds that could present similar symptoms. Did the authors tried to assess which symptoms were present? If this symptoms' declaration is not more specific, it should be withdrawn from the survey.
4) I have a concern with the characteristics of investigated people. Only age, sex, level of education and marital status are recorded. It is also important to know if people are in retirement home or their own place. For example many of the over 80 years old are in retirement home whether the younger ones remain in their home.
5) It is surprising that infection with Covid 19 is not associated with comorbidity such as diabetes or hypertension. It may depend on the level of blood pressure as well as the intensity of diabetes. We do not have information on the validity and/or intensity of these diseases. Were they the opinions o interviewed persons or recorded pathologies controlled by a medical doctor?
6) The outcome of Covid 19 is not given. It can be asymptomatic or higly pathogenic. I suppose that someone presenting a dangerous disease is more prone to be anxious than one having few light symptoms. Among the interviewed people were they cases of people with heavy symptoms? If they were with negligible pathologies it may explain that comorbidity were not important?
Some minor problems:
The LOST is included in the title and we need to reach Materials and methods to know what it is. It is not even in the summary.
l 78: health instead of heath
Provinces are with abbreviations in Table 1 and we may not know what theuy are. I also think there should be some explanations on the reasons to take into account provinces.
Again LOST in a table without full meaning (LOck..etc)
I think that a major revision is needed.
Author Response
Point 1: 4400 persons are included and were contacted by phone. I guess that quite a few did not respond to the questions for different reasons. What was the percentage of persons not answering to the questions?
Response 1: We added the details on the response rate to the second paragraph of the Methods section.
Point 2: The PCR on swab is associated with recent infection, the immunological test with contact with the virus from some days to several weeks ago. Concerning anxiety or depression the timing of infection may be very important: I am anxious because I just got it and I am depressed because I got it weeks ago, since depression does not appear in few days. Even for other characteristics (age, marital status, etc..) the fact that you just become infected, or you were infected may yeld different ORs. I suggest that the analysis should be done on swab and immunological tests separately.
Response 2: We thank the Reviewer for giving us the opportunity to better clarify an important point. As we now specified in the Methods section, participants were asked to report if they had been infected by SARS-CoV-2 since the beginning of the pandemic. Therefore, we do not have information on the timing of infection. This does not allow us to analyze the relationship with mental health outcomes according to the time since infection. This has been added to the limitations in the Discussion section.
Point 3: A good part of the 4400 people declared having symptoms of Covid 19. I wonder if at this period flu was present or changes of temperature caused colds that could present similar symptoms. Did the authors tried to assess which symptoms were present? If this symptoms' declaration is not more specific, it should be withdrawn from the survey.
Response 3: Here we report the translated questions from the questionnaire:
E.1) Do you currently have COVID-19 or have you ever had it?
â–¡ Yes
â–¡ No
E.2) If you answered “Yes” to the previous question:
â–¡ COVID-19 was diagnosed through a positive swab
â–¡ COVID-19 was diagnosed through a positive serological test
â–¡ I did not have a diagnosis, but I had evident symptoms of COVID-19
In the Methods section, we better explained that we asked participants if they had experienced evident symptoms of COVID-19.
Actually, the contribution of this self-reported information was not influential on the overall findings or associations found. In fact, conducting a sensitivity analysis in which we only considered subjects with a positive swab or serological test (see Table 1 and Table 2), the main findings did not change substantially.
We prefer therefore to keep the self-reported contribution. In fact, as already mentioned in the main text, since the incidence of many respiratory diseases dropped in 2020 (Yum et al. 2021), it can be expected that a large proportion of the patients who reported COVID-like symptoms had actually contracted a SARS-COV-2 infection. Given the difficulty in receiving an instrumental diagnosis in the first wave, we thought it was interesting to report this finding, which indicates that the real incidence of SARS-CoV-2 infection was underestimated, as confirmed by a large seroprevalence study carried out in the same period (Istituto Nazionale di Statistica; Italian Ministry of Health 2020).
Point 4: I have a concern with the characteristics of investigated people. Only age, sex, level of education and marital status are recorded. It is also important to know if people are in retirement home or their own place. For example, many of the over 80 years old are in retirement home whether the younger ones remain in their home.
Response 4: We agree with the Reviewer that this is an important point. In the last sentence of the Introduction section, we specified that the sample included only “home-dwelling older adults”. Now we replaced “home-dwelling” with “community-dwelling” in the Introduction section and we also specified this in the Methods section.
Point 5: It is surprising that infection with Covid 19 is not associated with comorbidities such as diabetes or hypertension. It may depend on the level of blood pressure as well as the intensity of diabetes. We do not have information on the validity and/or intensity of these diseases. Were they the opinions of interviewed persons or recorded pathologies controlled by a medical doctor?
Response 5: As we performed a telephone-based cross-sectional survey, the presence of a set of selected major chronic conditions was self-reported by the participants. However, we asked for the presence of a diagnosis of such conditions. This has been better specified in the Methods section. Thus, we are confident in the validity of the diagnoses.
Our interpretation is that chronic patients effectively protected themselves from the infection since they were more likely aware that they would develop more severe forms of the disease in case of SARS-CoV-2 infection.
Point 6: The outcome of Covid 19 is not given. It can be asymptomatic or higly pathogenic. I suppose that someone presenting a dangerous disease is more prone to be anxious than one having few light symptoms. Among the interviewed people were they cases of people with heavy symptoms? If they were with negligible pathologies, it may explain that comorbidity were not important?
Response 6: As mentioned above, the questionnaire did not include questions on the severity of the disease. We added this point to the study’s limitations.
Minor point 1: The LOST is included in the title and we need to reach Materials and methods to know what it is. It is not even in the summary.
Response to minor point 1: Added to the Introduction section
Minor point 2: l 78: health instead of heath
Response to minor point 2: Done
Minor point 3: Provinces are with abbreviations in Table 1 and we may not know what they are. I also think there should be some explanations on the reasons to take into account provinces.
Response to minor point 3: Thank you. We described them in the legend below the table.
Minor point 4: Again LOST in a table without full meaning (LOck..etc)
Response to minor point 4: Done
Reviewer 2 Report
I was invited to revise the paper entitled "Determinants of SARS-CoV-2 infection in the older adult population: Data from the LOST in Lombardia study". It was a cross-sectional study that aimed to estimate the prevalence of SARS-CoV-2 among Italian subjcets aged ove 65yo, andf to investigate associated factors. The study was conducted in Lombardy, a Northern Italian Region among the most affected during the first wave of pandemic.
I want to congratulate with Authors for the work. The paper is well written and easy to read.
The introduction was short and directly focused on the study background.
Methods section properly described the study design, the enrollment procedure and statistical analysis.
Results were clearly presented and were easy to read.
Discussion section was well structured and deeply explaine study results.
Minor observation:
- About study limitation, the prevalence estimation considereded only patients survived to the infection. Patients death after infection frequently have worse basal clinical condition compared to healed patients. In my opinion this point can be added to discussions.
Author Response
Point 1: About study limitation, the prevalence estimation considereded only patients survived to the infection. Patients death after infection frequently have worse basal clinical condition compared to healed patients. In my opinion this point can be added to discussions.
Response 1: Thank you for your thoughtful comment. We agree with the Reviewer and we added to the Discussion section the following sentence:
“It must be also taken into account that patients who die from COVID-19 frequently have worse basal clinical conditions compared to healed patients”
Round 2
Reviewer 1 Report
The authors have answered to the main queries and highligthted some limitations of their study. It can be published.